# VirDA: Reusing Backbone for Unsupervised Domain Adaptation with Visual Reprogramming

**Duy Nguyen**
*Hanoi University of Science and Technology*                    *duy.nd223435@hust.edu.vn*

**Dat Nguyen**
*Harvard University*                    *datnguyen@seas.harvard.edu*
*Basis.ai*                    *dat@basis.ai*

**Reviewed on OpenReview:** *https://openreview.net/forum?id=Qh7or7JRFI*

## Abstract

Image classification is among the pillars of computer-vision pipelines. While state-of-the-art models excel within their training domains, their performance often deteriorates when transferred to a new, unlabeled setting. Unsupervised domain adaptation (UDA) addresses this challenge by repurposing a well-trained source classifier for the target domain, enabling strong downstream results without the need for additional labeled data. Existing UDA pipelines fine-tune already well-trained backbone parameters for every new source-and-target pair, resulting in the number of training parameters and storage memory growing linearly with each new pair, and also preventing the reuse of these well-trained backbone parameters.

Inspired by recent implications that existing backbones have textural biases, we propose making use of domain-specific textural bias for domain adaptation via visual reprogramming, namely VIRDA. Instead of fine-tuning the full backbone, VIRDA prepends a domain-specific visual reprogramming layer to the backbone. This layer produces visual prompts that act as an added textural bias to the input image, adapting its "style" to a target domain. To optimize these visual reprogramming layers, we use multiple objective functions that optimize the intra- and inter-domain distribution differences when domain-adapting visual prompts are applied. This process does not require modifying the backbone parameters, allowing the same backbone to be reused across different domains.

We evaluate VIRDA on Office-31 and obtain 92.8% mean accuracy with only 1.5M trainable parameters. VIRDA surpasses PDA, the state-of-the-art parameter-efficient UDA baseline, by +1.6% accuracy while using just 46% of its parameters. Compared with full-backbone fine-tuning, VIRDA outperforms CDTrans and FixBi by +0.2% and +1.4%, respectively, while requiring only 1.7% and 2.8% of their trainable parameters. Relative to the strongest current methods (PMTrans and TVT), VIRDA uses 1.7% of their parameters and trades off only 2.2% and 1.1% accuracy, respectively[1].

## 1 Introduction

Recent advancements in image classification have significantly enhanced model performance through supervised learning, driven primarily by large amounts of labeled data (He et al., 2016; Dosovitskiy et al., 2020; Liu et al., 2021; Han et al., 2022). However, these supervised approaches struggle when applied to new, unlabeled domains due to domain shifts (Ganin & Lempitsky, 2015). This challenge is particularly prominent in fields involving emerging technologies, such as medical imaging for newly discovered diseases, where acquiring labeled data is costly and time-consuming (Abedi et al., 2024).

---

[1]We release our implementation and reproduction package at `https://github.com/Duy-Nguyen-Duc/VirDA`

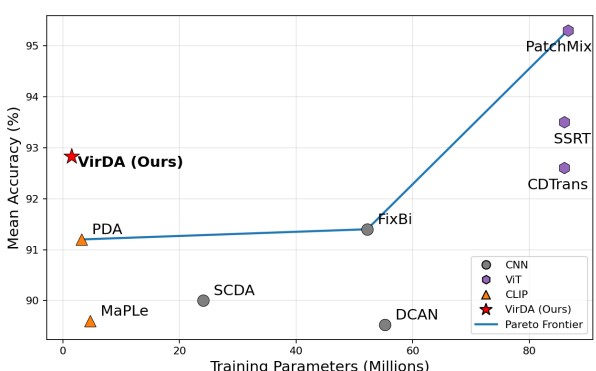

(a) The Pareto chart of existing methods, with VɪʀDA displaying the trade-off between accuracy and the number of training parameters.

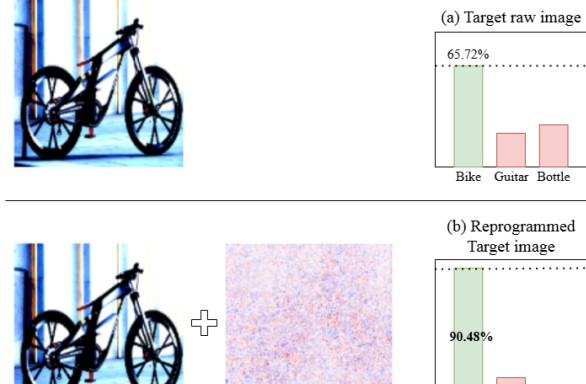

(b) Classification result of a well-trained model on the source domain enhances when applied to the target domain, due to using a pixel-wise textural mask.

Figure 1: Notably, our method excels over other parameter-efficient fine-tuning methods using CLIP as the backbone (e.g., PDA and MaPLe), as well as other methods that require full fine-tuning (e.g., FixBi and CDTrans) at minimal computation cost. Moreover, VɪʀDA required only 1.7% training parameters (1.5M to 86.6M) while sacrificing 2.2% accuracy compared to the SoTA method.

To address this limitation, Unsupervised Domain Adaptation (UDA) (Ganin & Lempitsky, 2015; Ganin et al., 2016; Saito et al., 2018), aims to adapt well-trained source domain classifiers to the target domains, given that no labels of the target domain are available. Prior UDA methods aim to transfer well-learned representations, the hidden features that are shareable between the source and target domains, and are invariant to the domain-specific style (e.g., differences in studio-lighting versus practical lighting condition (Venkateswara et al., 2017), coloring style between synthetic versus real-world imagery (Saenko et al., 2010)). Existing works (Zhang et al., 2019; Yang et al., 2023; Na et al., 2021) facilitate this transfer by adapting the hidden features produced by the backbone in a typical image classification framework (Deng et al., 2009; Krizhevsky et al., 2012) through various means, e.g., feature distribution alignment (Chen et al., 2019a;b; Sun et al., 2017) and adversarial training (Long et al., 2018; Ganin & Lempitsky, 2015). Others utilize the pretrained Vision-Language models to use the common textual label set to guide the adaptation (Ge et al., 2022; Du et al., 2024; Bai et al., 2024).

While these methods work well and achieve high accuracies, the transfer requires fine-tuning both the well-trained classifier along with fine-tuning the backbone, either a convolutional neural network (CNNs) (Long et al., 2018; Ganin & Lempitsky, 2015) or Vision Transformer backbones (Yang et al., 2023; Zhu et al., 2023; Xu et al., 2022; Liang et al., 2020), for each new domain. This limits backbone reusability, as well as requiring a large amount of storage to store the trained backbone across different source-target domain pairs.

In this paper, we propose Vɪsually Rᴇprogrammed Domain Adaptation (VɪʀDA), which aims to facilitate backbone reuses through a lightweight visual reprogramming layer. Recent findings show that even well-trained backbones that produce robust features still have textural bias, making them overly reliant on textural patterns for each domain and each category inside the domain (Geirhos et al., 2018). We thus exploit these textural biases for domain adaptation through visual reprogramming (Cai et al., 2024b): each visual reprogramming layer consists of a domain-specific textural pattern that aims to capture the domain-specific textural bias, and a per-instance mask generator that adaptively applies this textural pattern over the input image. Applying this visual reprogramming layer is thus equivalent to shifting the style of the input image (either from the source or the target domain) towards a common style, learned by the backbone (Figure 1b for an illustration). More specifically, this visual reprogramming is prepended to the backbone and thus does not require backbone modification or fine-tuning. Thus, our resulting architecture for each domain is composed of three modules in a cascaded manner: (1) a domain-specific visual reprogramming

layer, (2) a frozen, reusable backbone, and (3) a domain-specific classifier. To perform UDA and to train these visual-reprogramming layers and domain-specific classifiers, we design two key objectives: (1) an inter-domain alignment objective that aligns the learned hidden features and classification uncertainty from both the source and target domains, and (2) an intra-domain alignment objective that aims to learn domain-specific features through self-supervised loss.

We conduct experiments to evaluate VIRDA's capability in classification effectiveness, training parameter size, and storage requirement for each source and target domain pair. Our experiments demonstrate that the proposed VIRDA, requiring only a maximum of 1.5 million of training parameters (less than 2% of PMTrans (Zhu et al., 2023)) and only 6 MB for storage per domain (compared to over 340 MB of both PMTrans and CDTrans (Xu et al., 2022)), and fully reusing the backbone's parameters, achieves comparable performance to state-of-the-art (SOTA) methods across standard domain adaptation benchmarks, including Office-31 (Saenko et al., 2010), Office-Home (Venkateswara et al., 2017), and Digits (MNIST (LeCun et al., 1998), USPS (Netzer et al., 2011), SVHN (Hull, 1994)). The main contributions of this paper are summarized as follows.

- We propose a novel UDA method that efficiently addresses domain shifts by exploiting inherent textural biases in pretrained models, enabling lightweight yet effective domain adaptation.

- To the best of our knowledge, we are the first to perform domain adaptation with an entire single-modality frozen backbone by integrating visual reprogramming within the framework.

- We evaluate our method on three widely used benchmarks, confirming the effectiveness of our method with only a fraction of the training parameters, while achieving competitive performance compared to existing methods (as shown in Figure 1a).

The remaining of this paper is structured as follows: Section 2 reviews related works to ours; Section 3 presents the detailed architecture and methodology of VIRDA; Section 4 provides comprehensive experimental results and extensive ablation studies; and finally, Section 5 summarizes our conclusion as well as future research directions.

## 2 Related Works

In this section, we summarize three related research directions to VIRDA, namely, unsupervised domain adaptation, parameter-efficient fine-tuning for domain adaptation, and visual reprogramming.

**Unsupervised Domain Adaptation** As introduced in Section 1, existing UDA methods aim to align the well-learned hidden representation of the backbone's output. (Long et al., 2015) and (Long et al., 2017) proposes Deep Adaptation Network (DAN) and Joint Adaptation Network (JAN) that align different task-specific hidden representations by aligning their embedding distances with Maximum Mean Discrepancy (MMD), (Wen et al., 2019) instead uses uncertainty matching to align hidden features. (Ganin & Lempitsky, 2015), on the other hand, performs feature alignments through the use of inverse gradient from an adversarial domain discriminator, aiming to make the source and target features indistinguishable. (Long et al., 2018) improves per-class feature representation by clustering hidden features for adversarial training. (Tzeng et al., 2017) combines both feature distribution alignment and adversarial training. (Saito et al., 2018) and (Li et al., 2020) aim to further improve the precision of feature alignment using the classifier's disagreement to pinpoint and domain-specific attention modules (Saito et al., 2018; Li et al., 2020). Other works employ self-ensembling frameworks, improving consistency of predictions under perturbations with contrastive losses and self-supervised losses (Cui et al., 2020b; Yue et al., 2021; Xu et al., 2022), leveraging temporal smoothing to stabilize representations (Tarvainen & Valpola, 2017), and adapting Batch Nuclear-norm Maximization on the output matrix to improve prediction results (Cui et al., 2020a). Finally, recent methods combine both inter- and intra-domain alignment strategies (Na et al., 2021; Yang et al., 2023). We adapt these alignment strategies, specifically, we perform inter-domain alignments on the visual reprogramming layers with uncertainty matching and adversarial training, as well as intra-domain alignment with self-supervised loss and consistency objectives.

**Parameter-efficient fine-tuning (PEFT) for domain adaptation** As an alternative to fully fine-tuning the backbone, recent PEFT approaches leverage large vision–language models' multimodal inference capability and instead learn a small number of prompt or fine-tuning image-text adapter modules. MaPLe (Khattak et al., 2023) injects and finetunes a small set of context prompting tokens for each text-encoding layer to better align the hidden representation of domain-specific visual and text-based tokens. DAPL (Ge et al., 2022) also aims to modify CLIP for UDA, however, they finetune class-aware and domain-aware textual prompts with pseudo-labels. PDA (Bai et al., 2024) aims to better learn cross-domain shift by combining learned prompts with a lightweight image-guided feature tuning branch that performs distribution alignment under pseudo-labels. DAMP (Du et al., 2024) pushed the idea further by replacing image-guided feature tuning with a more powerful multi-domain transformer decoder. Our method can also be seen as fine-tuning "prompts" as a means to perform UDA. However, instead of learning a textual prompt that applies to a large pretrained multimodal model, we apply the visual prompt to highlight that domain adaptation is feasible via lightweight visual reprogramming of the input/patch-embedding space, where a small set of learnable image-side tokens steers a (mostly) frozen backbone-achieving parameter-efficient, backbone-agnostic adaptation. Because our approach does not make the assumption of using a multi-modal vision-language model, it can be applied to any of the existing backbones, i.e., both convolutional neural network-based backbone or vision transformer-based backbone.

**Visual Reprogramming** Visual Reprogramming (VR) is a method that repurposes pretrained vision backbones by learning small input-side modifications (e.g., additive prompts or adversarial "programs") so that the fixed or lightly adapted model performs a new downstream task without full retraining (Cai et al., 2024b). Recent works in VR focusing on learning perturbations guided by descriptive and distinctive attributes to improve alignment in Vision-Language Model (Cai et al., 2025), or incorporating adversarial examples to improve the robustness of re-programmed models (Zhou et al., 2025). However, these methods are only applicable to supervised training, and leave a research gap when applied to the unsupervised characteristics of UDA. Other visual prompting techniques have been applied to UDA, for example, in image classification (Gao et al., 2022), or image segmentation (Ma et al., 2023), employing visual prompting inside transformer architectures by inserting prompts into intermediate layers to facilitate cross-domain alignment. While VIRDA also uses visual reprogramming for UDA, we aim to leave the backbone unmodified to facilitate its reuse; instead, we rely on the aforementioned domain alignment objectives to train the visual reprogramming layers.

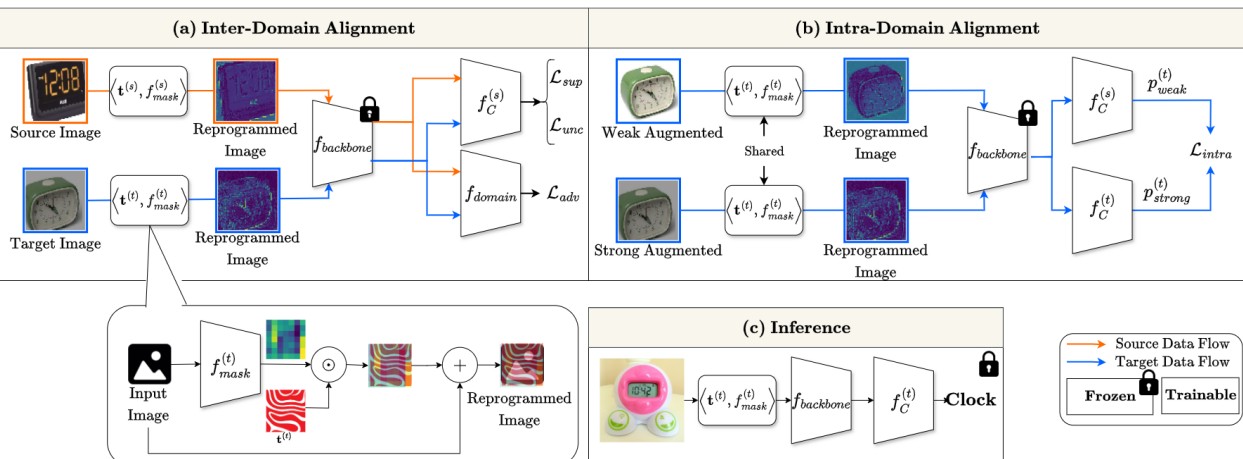

Figure 2: The overall pipeline of VIRDA.

## 3 Methodology

The unsupervised domain adaptation (UDA) problem for image classification (Ganin & Lempitsky, 2015) takes as input two datasets: a labeled dataset $\mathcal{D}_S = \{(\mathbf{x}_i^{(s)}, \mathbf{y}_i^{(s)})\}_{i=1}^{N_s}$ from a source domain $\mathcal{S}$ along with an

unlabeled target domain $\mathcal{T}$'s dataset $\mathcal{D}_T = \{\mathbf{x}_j^{(t)}\}_{j=1}^{N_t}$, where each $\mathbf{x}^{(s)}$ and $\mathbf{x}^{(t)}$ is the input data samples (i.e., images) from the source and target domain respectively, and $\mathbf{y}^{(s)}$ is the sample label (i.e., image labels). We denote the mini-batch of input images be $\mathbf{B}^{(s)} = \{\mathbf{x}_1^{(s)}, \mathbf{x}_2^{(s)}, \ldots, \mathbf{x}_k^{(s)}\}$ and $\mathbf{B}^{(t)} = \{\mathbf{x}_1^{(t)}, \mathbf{x}_2^{(t)}, \ldots, \mathbf{x}_k^{(t)}\}$ for the source and target domains, respectively, with the $y^{(s)} = \{\mathbf{y}_1^{(s)}, \mathbf{y}_2^{(s)}, \ldots, \mathbf{y}_k^{(s)}\}$ be the corresponding source labels. Although the unsupervised setting of the target domain $\mathcal{T}$, we assume that the two domains share an identical label space $\mathcal{Y}$. As previously described, VIRDA consists of a visual reprogramming layer that produces visual prompts describing textural and spatial shifts between different domains. We describe this representation briefly in Section 3.1.

To optimize this visual reprogramming layer, we follow the formulation of existing works and attempt to align the hidden features produced by applying these layers in Section 3.2. (See Figure 2). These hidden features are aligned intra-domain with domain-specific data augmentations, and inter-domain by aligning hidden features produced with different domain-specific visual reprogramming layers. Specifically, the inter-domain alignment loss is implemented using three sub-loss functions: the supervised source loss $\mathcal{L}_{sup}$, the adversarial loss that aims to align inter-domain feature alignment $\mathcal{L}_{adv}$, and $\mathcal{L}_{unc}$ that aligns between the class-wise prediction uncertainty of source-and-target domains. The intra-domain loss, meanwhile, is constructed using two sub-loss functions: (1) $\mathcal{L}_{unsup}$, an unsupervised consistency loss enforcing the same image under two different augmentations yields the same output, and (2) a confidence-distribution matching loss $\mathcal{L}_{distrib}$ that encourages the output confidence distributions from differently augmented views of the same image to be similar.

### 3.1 Encoding Domain-specific Textural and Transformational Visual Prompt

**Visual Prompt Representation** Assuming the input image is represented as a tensor $\mathbf{x} \in \mathbb{R}^{w \times h \times c}$, where $c$ is the number of color channels (usually 3), a visual reprogramming module is a pair $\langle \mathbf{t}, f_{mask} \rangle$, where $\mathbf{t} \in \mathrm{R}^{w \times h \times c}$ is the domain-specific textural pattern, while $f_{mask} : \mathbb{R}^{w \times h \times c} \to [0, 1]^{w \times h \times c}$ is the mask producer.

**Visual mask-producing Layer** A mask-producing layer $f_{mask}$ is a function that takes as input an image and produces the mask. The function is a fully convolutional subnetwork of $L_{vr}$ layers, where each layer $l \in \{1, ..., L_{vr}\}$ performs a $3 \times 3$ convolution with padding of 1. The feature map is then downsampled into non-overlapping patches of size $2^{N_{vr}} \times 2^{N_{vr}}$, allowing the network to learn a compact summary of local texture and shape cues within each patch. In our experiments, we follow Cai et al. (2024b) to set $L_{vr} \in \{5, 6\}$, while $N_{vr} \in [1, 5]$. While deeper reprogramming layers with larger patches excel at capturing coarse, object-level patterns, shallower layers with smaller patches preserve finer, more detailed features.

**Structural mask-producing layer.** Although a visual reprogramming layer can mask visual appearance shifts, the cross-domain mismatches are also displayed in spatial layout and feature dependency, so that the downstream model still learns the wrong structural priors. For example, product images often center the object, whereas real-world images place it off-center. To model this structural shift, we adopt the Coordinate Attention (CA) mechanism of Hou et al. (2021) to generate two axis-wise, position-sensitive masks that softly recenter attention. CA first aggregates features along one spatial axis at a time to preserve positional information along the other axis:

$$\mathbf{g}_h(i, 1, c) = \frac{1}{w} \sum_{j=1}^{w} \mathbf{x}(i, j, c) \in \mathbb{R}^{h \times 1 \times c}, \qquad \mathbf{g}_w(1, j, c) = \frac{1}{h} \sum_{i=1}^{h} \mathbf{x}(i, j, c) \in \mathbb{R}^{1 \times w \times c}. \tag{1}$$

The two aggregated tensors are concatenated and passed through a shared $1 \times 1$ convolution:

$$\mathbf{z} = \mathrm{Conv}_{1 \times 1}([\mathbf{g}_h, \mathbf{g}_w]) \in \mathbb{R}^{(h+w) \times 1 \times c}. \tag{2}$$

We then split $\mathbf{z}$ into $\mathbf{z}_h \in \mathbb{R}^{h \times 1 \times c}$ and $\mathbf{z}_w \in \mathbb{R}^{1 \times w \times c}$, and transform each branch with its own $1 \times 1$ convolution followed by a sigmoid gate $\sigma$ to obtain the axis-wise attention masks:

$$\mathbf{A_h} = \sigma(\mathrm{Conv}_{1 \times 1}^{(h)}(\mathbf{z}_h)) \in \mathbb{R}^{h \times 1 \times c}, \qquad \mathbf{A_w} = \sigma(\mathrm{Conv}_{1 \times 1}^{(w)}(\mathbf{z}_w)) \in \mathbb{R}^{1 \times w \times c}. \tag{3}$$

Broadcasting $\mathbf{A_h}$ and $\mathbf{A_w}$ over the missing dimension yields our position-shifting masks (vertical and horizontal, respectively). Applying them reweights rows and columns-analogous to softly shifting emphasis toward locations that are more likely in the target domain:

$$f_{\text{coord}}(\mathbf{x}) = \mathbf{x} \odot \mathbf{A_h} \odot \mathbf{A_w}. \tag{4}$$

where $\odot$ is the element-wise multiplication operation. We optimize the parameters of this structural layer jointly with the visual reprogramming objectives described in Sec. 3.2.

**Visual reprogramming layer** The visual re-programming layer $f_{pre}$ works as follows:

$$f_{pre}(\mathbf{x}) = f_{coord}(\mathbf{x}) + \mathbf{t} \odot f_{mask}(f_{coord}(\mathbf{x})) \tag{5}$$

The textural pattern $\mathbf{t}$ and both the mask-producing layers $f_{mask}, f_{coord}$ are domain-specific. Hence, for each domain $d \in \{s, t\}$ we denote $\mathbf{t}^{(d)}$, $f_{mask}^{(d)}$ and $f_{coord}^{(d)}$ for domain $d$'s pattern and mask-producing layers, and $f_{pre}^{(d)}$ as the visual reprogramming layer for domain $d$.

Intuitively, each visual prompt is a combination of structural masking and textural reprogramming. Hypothetically, the structural mask, i.e., the successive application of $f_{coord}$ and $f_{mask}$, highlights the regions to be re-programmed for each image, while the textural bias should capture the domain-specific style that the backbone has biases towards, following the finding of the existing work (Geirhos et al., 2018).

## 3.2 Visually-aligning Model

Unlike previous visual reprogramming works (Cai et al., 2024b;a; Zhou et al., 2025; Cai et al., 2025) that assume the availability of target-domain labels, our approach must function in an unsupervised setting. Consequently, our classifier architecture requires sufficient flexibility to leverage both shared (domain-invariant) and unique (domain-specific) features effectively (Xiao et al., 2021). To achieve this, we introduce a visual reprogramming module that explicitly couples domain-invariant features, combined with loosely coupled, domain-specific classifier heads that capture specialized features.

**Full model architecture** Thus, we design an architecture that couples the common features through a visual-reprogramming module, while also enabling learning to make use of domain-specific features, through lowly-coupled domain-specific classifiers. Following Eq. 5, we denote the domain-specific visual-reprogramming module for the source domain $s$ and the target domain $t$ as $f_{pre}^{(s)}$ and $f_{pre}^{(t)}$, respectively. We also denote the domain-specific classifiers for the source and target domains as $f_C^{(s)}$ and $f_C^{(t)}$, respectively. Thus, our model architecture is as depicted in Figure 2: for each domain, we cascade three modules: a domain-specific visual-reprogramming module $f_{pre}$, a reused domain-invariant backbone $f_{backbone}$, and finally a domain-specific classifier $f_C$. To train this architecture, we have to model two goals: (1) we have to perform inter-domain alignment, and (2) we have to perform intra-domain alignment.

**Inter-domain alignment** Since we do not re-train our backbone, as such, inter-domain alignment implies different objectives for the visual reprogramming modules and the domain-specific classifier, respectively. For the source and target visual re-programming modules $f_{pre}^{(s)}$ and $f_{pre}^{(t)}$, this alignment means that these modules have to be able to "shift" the input image style from their respective domain to that of a common image style that was learned by the shared backbone. For the domain-specific classifiers $f_C^{(s)}$ and $f_C^{(t)}$, this means that they have to learn to use the shared or aligned inter-domain features. Concretely, the inter-domain loss is implemented via three objectives:

$$\mathcal{L}_{inter} = \mathcal{L}_{sup} + \mathcal{L}_{adv} + \mathcal{L}_{unc} \tag{6}$$

To keep the source classifier well-trained during adaptation, which is crucial to transfer the feature-label information when $f_{pre}^{(s)}$ learned the common style, we first implement supervised loss on the source data:

$$\mathcal{L}_{sup} = \frac{1}{k} \sum_{i=1}^{k} \mathbf{CE}(\mathbf{y}_i^{(s)}, p_i^{(s)}), \tag{7}$$

where $p_i^{(d)} = f_C^{(d)}(\mathbf{z}_i^{(d)})$ and $\mathbf{z}_i^{(d)} = \big(f_{backbone} \circ f_{pre}^{(d)}\big)(\mathbf{x}_i^{(d)})$ is the prediction and features obtained for each sample in the domain $d$'s batch $\mathbf{B}^{(d)}$, respectively. The $\mathcal{L}_{adv}$ is the adversarial domain discriminator loss, used to help the visual-reprogramming modules to shift their respective domain images into a common domain and produce indistinguishable hidden features. This loss is modeled via binary cross-entropy over the predictions produced by a domain-specific discriminator $f_{domain}$. Hypothetically, both the source- and target- visual reprogramming modules will produce aligned hidden features that can "fool" the domain discriminator $f_{domain}$, defined as:

$$\mathcal{L}_{adv} = \frac{1}{k} \sum_{i=1}^{k} \big[\log f_{domain}(\mathbf{z}_i^{(s)}) + \log\big(1 - f_{domain}(\mathbf{z}_i^{(t)})\big)\big] \tag{8}$$

If the domain discriminator can still distinguish the produced features, then the backpropagated signals will be used to train both $f_{domain}$ and the visual-reprogramming modules through reverse gradient (Ganin & Lempitsky, 2015). Given that the source and target domain hidden features are aligned, the source classifier $f_C^{(s)}$ should be generalized to both domains to leverage the use of labels. This means that it can both learn robust domain-invariant features produced by the target visual re-programming module $f_{pre}^{(t)}$, while remaining well-trained on source domain inputs. Following prior work (Wen et al., 2019), we propose to use the uncertainty loss $\mathcal{L}_{unc}$. This objective is implemented to align $f_C^{(s)}$'s uncertainty in different domains, i.e., the distribution of $f_C^{(s)}$'s uncertainty should be the same on both the source and target-reprogrammed domain. Indirectly, this objective will also enforce inter-domain alignment between two visual-reprogramming modules. Recall that we have a batch of input images from both domains, $\mathbf{B}^{(s)}$ and $\mathbf{B}^{(t)}$. We perform $M$ stochastic forward passes of the full module and sample the per-instance, per-class uncertainty on both the reprogrammed source batch $\hat{\mathbf{B}}^{(s)}$ and the reprogrammed target batch $\hat{\mathbf{B}}^{(t)}$. For each instance $i$ and class $c$, we collect the $M$ predictive samples $\{p_{m,c}(y \mid \mathbf{x}_i)\}_{m=1}^{M}$ and treat them as draws from a per-instance, per-class uncertainty distribution. We use these point-wise uncertainty estimation to estimate the full uncertainty distributions $\hat{q}_{i,c}^{(s)}$ and $\hat{q}_{i,c}^{(t)}$. We then align uncertainties by minimizing the class-averaged KL divergence:

$$\mathcal{L}_{unc} = \frac{1}{k\,|\mathcal{Y}|} \sum_{i=1}^{k} \sum_{c \in \mathcal{Y}} \mathbb{KL}\big(\hat{q}_{i,c}^{(s)} \,\|\, \hat{q}_{i,c}^{(t)}\big), \tag{9}$$

This encourages matched predictive uncertainty across domains and indirectly aligns the two visual-reprogramming modules. In practice, we place Dropout layers with different probabilities $p_{mask}$ and $p_C$ in both $f_{mask}^{(d)}$ and $f_C^{(s)}$, respectively to measure uncertainty (Gal & Ghahramani, 2016).

**Intra-domain alignment** To enhance the performance of our model on the target domain, we employ intra-domain objectives to transfer the already-well-trained source classifier $f_C^{(s)}$'s classification capability to the target domain classifier $f_C^{(t)}$, while also allowing the target domain classifier to learn intra-domain robust feature. Specifically, the intra-domain alignment loss is:

$$\mathcal{L}_{intra} = \mathcal{L}_{unsup} + \mathcal{L}_{distrib} \tag{10}$$

where both of our objectives enforce consistency through different views of a sample. This is done by augmenting the target domain image through both strong augmentations, such as affine transformation or color jitter, and weak augmentations, which are the normalization of original image. The resulting augmented images' hidden features are then passed through both the source classifier $f_C^{(s)}$ and the target classifier $f_C^{(t)}$. Assuming the features are well aligned to a certain level through prior objectives, we use $f_C^{(s)}$ to produce pseudo labels for the target classifier $f_C^{(t)}$. The $\mathcal{L}_{unsup}$ is thus to optimize $f_C^{(t)}$ to minimize both the difference in prediction of $f_C^{(t)}$ and $f_C^{(s)}$, as well as reducing $f_C^{(t)}$'s uncertainty over augmented target images while the $\mathcal{L}_{distrib}$ penalises any disagreement between the target classifier $f_C^{(t)}$ and the source classifier $f_C^{(s)}$. More concretely, let $f_{weak}$ denote the weak augmentation and $f_{strong}$ denote the strong augmentation. From $\mathbf{B}^{(t)}$, we obtain the weak-view prediction $p_{i,weak}^{(t)} = \big(f_C^{(s)} \circ f_{backbone} \circ f_{pre}^{(t)} \circ f_{weak}\big)(\mathbf{x}_i^{(t)})$ and the strong-view

prediction $p_{i,strong}^{(t)} = \left(f_C^{(t)} \circ f_{backbone} \circ f_{pre}^{(t)} \circ f_{strong}\right)(\mathbf{x}_i^{(t)})$. Thus, the distribution divergence loss is:

$$\mathcal{L}_{distrib} = \frac{1}{k}\sum_{i=1}^{k}\mathbb{KL}(p_{i,weak}^{(t)} \,\|\, p_{i,strong}^{(t)}) \tag{11}$$

We use the most confident predictions from the weak augmentation branch for each sample, $\hat{y}_i^{(t)} = \arg\max p_{i,weak}^{(t)}$, to construct our unsupervised loss:

$$\mathcal{L}_{unsup} = \frac{1}{k}\sum_{i=1}^{k}\mathbf{CE}(\hat{y}_i^{(t)}, p_{i,strong}^{(t)}) \tag{12}$$

### 3.3 Training objectives and inference

To train our model using the five losses, the training objectives can be defined as:

$$\mathcal{L} = \alpha_{unc} * \mathcal{L}_{unc} + \alpha_{adv} * \mathcal{L}_{adv} + \alpha_{intra} * \mathcal{L}_{intra} \tag{13}$$

At inference time, we perform on the target domain using both the target visual reprogramming layer and the target classifier for label prediction:

$$\hat{y}_i = \arg\max\left(f_C^{(t)} \circ f_{backbone} \circ f_{pre}^{(t)}(x_i)\right), \tag{14}$$

where $\hat{y}_i$ is the predicted class of the unlabeled target sample.

## 4 Experiments

We evaluate our proposed method on widely used three domain adaptation benchmarks, namely Office-31, Office-Home, and Digits, compared with state-of-the-art UDA methods in both accuracy and number of training parameters. In addition, we validate the contributions of the proposed method through extensive ablation studies. We describe detailed dataset characteristics and implementation details below.

### 4.1 Datasets

**Digits** is a dataset composed from three other digit datasets, which are MNIST (LeCun et al., 1998), USPS (Hull, 1994), and Street View House Numbers (SVHN) (Netzer et al., 2011). In terms of domain characteristics, MNIST (M) contains grayscale digit images with a clean background; SVHN (S) consists of cropped coloured digits from real scenes with extremely blurred appearance; USPS (U) provides grayscale handwritten digit images with unconstrained writing styles. Whilst sharing the same 10 (0-9) digit classes, the three datasets present cd different data distributions, therefore suitable for UDA evaluation. For the UDA test, we adopted three commonly used cross-dataset transfer settings with the standard data split: S→M, U→M, M→U.

**Office-31** (Saenko et al., 2010) is the most popular dataset for real-world domain adaptation. It contains 4,110 images of 31 categories in three domains: Amazon (A), Webcam (W), DSLR (D). We evaluated all methods on six domain adaptation tasks.

**Office-Home** (Venkateswara et al., 2017) is a more challenging benchmark than Office-31. It consists of images of everyday objects organized into four domains: artistic images (Ar), CLIP art (Cl), product images (Pr), and real-world images (Rw). It contains 15,500 images of 65 classes.

### 4.2 Implementation Details

In all experiments, we use both Resnet (He et al., 2016) and ViT (Dosovitskiy et al., 2020) models pre-trained on ImageNet (Deng et al., 2009) as the fixed backbone for VIRDA. For the Digits tasks, we use ResNet-18

with a learning rate of $3e^{-4}$ for the classifier heads and $5e^{-4}$ for the visual reprogramming modules, using a batch size of 128. The dropout rate for the classifier and the mask generator is set as $p_{mask} = 0.5$ and $p_C = 0.3$, respectively. On the Office-Home and Office-31 datasets, we adopt ViT-B/32 as the backbone for all transfer tasks. We set the same learning rate as above, using a batch size of 32 and $p_{mask} = 0.3$ with $p_C = 0.1$. For all experiments, we adopt AdamW (Loshchilov & Hutter, 2017) with the default configuration of $(\beta_1, \beta_2)$ is $(0.9, 0.999)$, and a weight decay of $1e^{-5}$. On the Office-31 and Office-Home datasets, we set $L_{vr}$ and $N_{vr}$ corresponding to 6 and 5, for coarse object-level mask, while on Digits we set $L_{vr} = 5$ and $N_{vr} = 4$, as the dataset's characteristics demonstrate mild transformation. On all tasks, we set the number of forward passes to estimate uncertainty $M = 4$. We train our method in two phases, namely burn-in in 20 epochs and domain adaptation in 30 epochs for all data settings following Li et al. (2022). The scaling factor for each loss is set as $\alpha_{unc} = 0.3$, $\alpha_{adv} = 0.1$ and $\alpha_{intra} = 0.15$, in which these hyperparameters are obtained via grid-search. We also provide the reproduction package to reproduce this grid-search and training process. The experiment results reported here were obtained on a machine equipped with Intel Xeon Gold 6130 CPU at 2.10GHz clock speed with 16 cores and 64 GB of RAM running Linux and using a single NVIDIA GTX 3090 device. We also provide a quantitative comparison in Table 7 showing the differences in training, memory, and computational efficiency between our approach and full-backbone fine-tuning.

### 4.3 Results

To provide comparison, we compare VIRDA with the widely-recognized state-of-the-art methods on Office-31 and Office-Home datasets that use different backbones. Specifically, we include MSTN (Xie et al., 2018), DCAN (Li et al., 2020), SCDA (Li et al., 2020), FixBi (Na et al., 2021) as baselines for the Resnet backbone; ViT-based, SSRT (Sun et al., 2022), PMTrans (Zhu et al., 2023), CDTrans (Xu et al., 2022), TVT (Yang et al., 2023) as the baselines for the ViT backbone; DAMP (Du et al., 2024), PDA (Bai et al., 2024), MaPLe (Khattak et al., 2023), DAPL (Ge et al., 2022) as the baselines for parameter-efficient fine-tuning on the CLIP backbone. These include a wide range of backbones, especially, with CLIP, VIRDA is a plugin to a backbone that can already incorporate text features. We follow the setting of Ge et al. (2022) and apply a VR layer priorly to the image encoder to report our model with the CLIP backbone. Note that on all of our transfer tasks, we keep the backbone frozen, and we estimate the number of training parameters of our method on the visual prompt layers and the classifier heads.

**Results on Digits.** We display the performance of VIRDA on Digits tasks in Tab. 1, where it achieves a mean accuracy of 94.8% on the benchmark using only 0.2M training parameters, which is better than Cy-CADA (Hoffman et al., 2018) and GTA (Sankaranarayanan et al., 2018), while underperforming MCD (Saito et al., 2018) by a small margin. Compared to DANN, which uses the most parameters in our benchmarks, VIRDA delivers higher accuracy on all tasks. While MCD achieves a slightly higher mean (95.6%), VIRDA remains competitive across all shifts while operating with the smallest model size among the compared methods.

Table 1: Accuracy (%) on Digits for UDA (ResNet backbone). We highlight the best results in **bold**, and the second best results in underscore.

| Method | Training Params (M) | M→U | U→M | S→M | Mean |
|---|---|---|---|---|---|
| DANN | 21.0 | 90.4 | 94.7 | 84.2 | 89.8 |
| CyCADA | 0.4 | 95.6 | **96.5** | 90.4 | 94.2 |
| GTA | 0.4 | 95.3 | 96.4 | 92.4 | 94.7 |
| MCD | 0.4 | **96.5** | 94.1 | **96.2** | **95.6** |
| **Ours (CNN)** | 0.2 | 95.4 | 95.9 | 93.0 | 94.8 |

**Results on Office-31.** As shown in Tab. 2, our method can improve the original backbone's fine-tuning methods by 2% to 10% and outperforms some of the prior methods, such as MSTN and CDTrans, while only requiring a fraction of training parameters. VIRDA is capable of achieving over 99.0% on mild shift tasks D↔W, while achieving strong results on tasks where the shift is hard but the source domain is visually more diverse than the target domain (such as A→W or A→D). With the CLIP baselines, we improve the baseline method DAPL by nearly 3% with the additional 0.5M training parameters, where on the mild shifts

(A→W, A→D and W↔ D) VIRDA consistently improves from 5% to 7%, while decreasing the accuracy on harder shift that requires semantic understanding, such as W→A or D→A.

Table 2: Accuracy (%) on Office-31 for UDA with ResNet, ViT, and CLIP backbones. We highlight the best results in **bold** and the second best results in underscore.

| Method | | Parameter size (M) | Training params (M) | A→W | D→W | W→D | A→D | D→A | W→A | Mean |
|---|---|---|---|---|---|---|---|---|---|---|
| Resnet50-based | ResNet | 23.8 | 23.8 | 68.6 | 96.8 | 99.3 | 69.1 | 62.8 | 61.0 | 76.1 |
| MSTN | | 59.24 | 59.2 | 91.3 | 98.9 | **100.0** | 90.4 | 72.7 | 65.6 | 86.5 |
| DCAN | | 55.2 | 55.2 | 95.0 | 97.5 | **100.0** | 92.6 | 77.2 | 74.9 | 89.5 |
| SCDA | | 24.0 | 24.0 | 94.2 | 98.7 | 99.8 | **95.2** | 75.7 | 76.2 | 90.0 |
| FixBi | | 52.2 | 52.2 | **96.1** | **99.3** | **100.0** | 95.0 | **78.7** | **79.4** | **91.4** |
| **Ours (CNN)** | | 25.6 | 2.1 | 84.9 | 96.2 | **100.0** | 90.2 | 64.2 | 66.7 | 83.7 |
| ViT-based | ViT | 86.0 | 86.0 | 91.2 | 99.2 | **100.0** | 90.4 | 81.1 | 80.6 | 90.4 |
| SSRT | | 86.0 | 86.0 | 97.7 | 99.2 | **100.0** | 98.6 | 83.5 | 82.2 | 93.5 |
| CDTrans | | 86.0 | 86.0 | 96.7 | 99.0 | **100.0** | 97.0 | 81.1 | 81.9 | 92.6 |
| TVT | | 86.0 | 86.0 | 96.4 | 99.4 | **100.0** | 96.4 | 84.9 | 86.0 | 93.9 |
| PMTrans | | 86.6 | 86.6 | **99.1** | **99.6** | **100.0** | **99.6** | **85.7** | **86.3** | **95.0** |
| **Ours (ViT)** | | 87.6 | 1.5 | 94.7 | 99.0 | 99.5 | 97.9 | 81.3 | 84.1 | 92.8 |
| CLIP-based (zero-shot) | CLIP | 124.0 | 0.0 | 75.8 | 75.8 | 77.7 | 77.7 | 79.0 | 79.0 | 77.5 |
| PDA | | 153.0 | 3.2 | **92.1** | **98.1** | **99.8** | **91.2** | **83.5** | **82.5** | **91.2** |
| MaPLE | | 154.4 | 4.7 | 88.6 | 97.7 | 99.4 | 86.9 | 83.0 | 82.0 | 89.6 |
| DAPL | | 124.3 | 0.3 | 80.3 | 81.8 | 81.8 | 81.3 | 81.2 | 81.0 | 81.2 |
| **Ours (DAPL)** | | 125.1 | 0.8 | 85.3 | 89.3 | 87.3 | 85.7 | 77.1 | 79.0 | 84.0 |

**Results on Office-Home.** VIRDA improves the baseline methods, such as CLIP and Resnet, by a minimum of 2%, while decreasing the accuracy when using the ViT backbones by nearly 1.5% with only a fraction of training parameters ranging from 0.8M to 2.1M of training parameters. With the ViT backbone, our method can improve and be on par with the full-finetuning baseline by 0.8% to 2.9% on mild shifts (e.g., Cl→Ar, Cl→Pr, and Pr↔Ar), while on the reverse shifts VIRDA constantly losing 3-5% in accuracy. However, the average accuracy of our methods when plugging in different backbones is still comparable with some of the prior methods, such as MSTN or CDTrans, while using only 1.7 to 3.5% of training parameters. Furthermore, when using the CLIP backbone, we outperform both the zero-shot CLIP and MaPLe in terms of accuracy, with only 0.8M of training parameters. We improve or are on par with the baseline method DAPL on 7 out of 12 tasks, with 0.3-0.5% accuracy gain, while losing at most 1.0% accuracy in the other 5 tasks, resulting in a slightly 0.1% mean accuracy overall. A detailed analysis of the behaviour of VIRDA is provided in Sec. 4.4.

Table 3: Accuracy (%) on Office-Home for UDA with ResNet, ViT, and CLIP backbones. We highlight the best results in **bold** and the second best results in underscore.

| Method | | Parameter size (M) | Training params (M) | Ar→Cl | Ar→Pr | Ar→Rw | Cl→Ar | Cl→Pr | Cl→Rw | Pr→Ar | Pr→Cl | Pr→Rw | Rw→Ar | Rw→Cl | Rw→Pr | Mean |
|---|---|---|---|---|---|---|---|---|---|---|---|---|---|---|---|---|
| Resnet50-based | ResNet | 23.8 | 23.8 | 34.9 | 50 | 58 | 37.4 | 41.9 | 46.2 | 38.5 | 31.2 | 60.4 | 53.9 | 41.2 | 59.9 | 46.1 |
| MSTN | | 59.24 | 59.2 | 49.8 | 70.3 | 76.3 | 60.4 | 68.5 | 69.6 | 61.4 | 48.9 | 75.7 | 70.9 | 55.0 | 81.1 | 65.7 |
| DCAN | | 55.2 | 55.2 | 54.5 | 75.7 | **81.2** | 67.4 | 74.0 | 76.3 | **67.4** | 52.7 | 80.6 | 74.1 | 59.1 | 83.5 | 70.5 |
| SCDA | | 24.0 | 24.0 | 57.5 | 76.9 | 80.3 | 65.7 | 74.9 | 74.5 | 65.5 | 53.6 | 79.8 | 74.5 | 59.6 | 83.7 | 70.5 |
| FixBi | | 52.2 | 52.2 | **58.1** | **77.3** | 80.4 | 67.7 | 79.5 | 78.1 | 65.8 | **57.9** | 81.7 | 76.4 | 62.9 | 86.7 | 72.7 |
| **Ours (CNN)** | | 25.6 | 2.1 | 49.4 | 70.1 | 76.0 | 61.3 | 70.0 | 71.5 | 61.6 | 46.4 | 77.6 | 68.6 | 54.0 | 79.2 | 65.5 |
| ViT-based | ViT | 86.0 | 86.0 | 67.0 | 85.7 | 88.1 | 80.1 | 84.1 | 86.7 | 79.5 | 67.0 | 89.4 | 83.6 | 70.2 | 91.2 | 81.1 |
| SSRT | | 86.0 | 86.0 | 75.2 | 89.0 | 91.1 | 85.1 | 88.3 | 89.9 | 85.0 | 74.2 | 91.2 | 85.7 | 78.6 | 91.8 | 85.4 |
| CDTrans | | 86.0 | 86.0 | 68.8 | 85.0 | 86.7 | 81.5 | 87.1 | 87.3 | 79.6 | 63.3 | 88.2 | 82.0 | 66.0 | 90.6 | 80.5 |
| TVT | | 86.0 | 86.0 | 74.9 | 86.8 | 89.5 | 82.8 | 88.0 | 88.3 | 79.8 | 71.9 | 90.1 | 85.5 | 74.6 | 90.6 | 83.6 |
| PMTrans | | 86.6 | 86.6 | **81.2** | **91.6** | **92.4** | **88.9** | **91.6** | **93.0** | **88.5** | **80.0** | **93.4** | **89.5** | **82.4** | **91.2** | **88.9** |
| **Ours (ViT)** | | 87.6 | 1.5 | 62.8 | 85.7 | 88.9 | 80.9 | 87.0 | 86.4 | 81.5 | 61.4 | 88.5 | 80.3 | 63.6 | 88.6 | 79.6 |
| CLIP-based (zero-shot) | CLIP | 124.0 | 0.0 | 67.6 | 89.0 | 89.4 | 82.4 | 89.0 | 89.4 | 82.4 | 67.6 | 89.4 | 82.4 | 67.6 | 89.0 | 82.1 |
| PDA | | 153.0 | 3.2 | 73.5 | 91.4 | 91.3 | 86.0 | 91.6 | 91.5 | 86.0 | 73.5 | 91.7 | **86.4** | 73.0 | 92.4 | 85.7 |
| MaPLe | | 154.4 | 4.7 | 72.2 | 91.6 | 90.3 | 82.6 | 90.9 | 89.8 | 82.4 | 71.6 | 90.1 | 85.1 | 72.0 | 92.1 | 84.2 |
| DAPL | | 124.3 | 0.3 | 70.7 | 91.0 | 90.9 | 85.2 | 91.0 | 90.9 | 85.1 | 70.7 | 90.9 | 85.3 | 70.4 | 91.4 | 84.5 |
| DAMP | | 131.1 | 6.7 | **75.7** | **94.2** | **92.0** | **86.3** | **94.2** | **91.9** | **86.2** | **76.3** | **92.4** | 86.1 | **75.6** | **94.0** | **87.1** |
| **Ours (DAPL)** | | 124.8 | 0.8 | 71.2 | 91.4 | 91.2 | 84.7 | 91.0 | 90.8 | 85.1 | 69.7 | 91.0 | 85.0 | 70.1 | 91.6 | 84.4 |

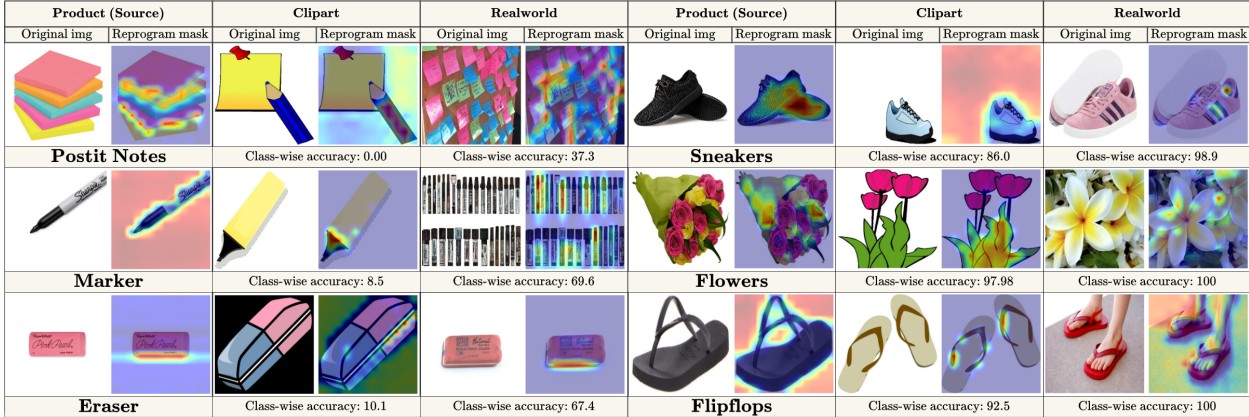

Figure 3: Samples and reprogram masks visualization of classes using VIRDA transfer from the source domain (Product) to the mild target domain (Realworld) and the hard target domain (Clipart). These classes are chosen from the classes where our method performs worst and best in Clipart, indicated by the class-wise accuracy of VIRDA.

Table 4: Ablation on Office-31 with ViT-B backbone finetuning. The best result are in **bold**.

| Finetuning layer | Training params (M) | D→A | W→A | A→W | Mean |
|---|---|---|---|---|---|
| Head | 1.1 | 72.6 | 74.6 | 88.9 | 78.7 |
| BB-Last | 8.2 | 72.1 | 79.1 | 87.1 | 79.7 |
| BB-Last-3 | 22.3 | 71.4 | 70.5 | 78.0 | 73.3 |
| BB-1 | 8.2 | 78.7 | 80.3 | 87.1 | 82.0 |
| **VirDA** | **1.5** | **81.3** | **84.1** | **94.7** | **86.7** |

## 4.4 Ablation studies

**Comparison with Partial Fine-Tuning.** While many UDA methods rely on full fine-tuning of both the backbone and the classifier, others achieve adaptation by updating only a subset of parameters. For example, TENT Wang et al. (2021) adapts to distribution shifts by optimizing only channel-wise affine transformations at test time. To evaluate how VIRDA compares with these approaches, we conduct experiments on the three most challenging Office-31 tasks (D→A, W→A, and A→W) using a ViT backbone Dosovitskiy et al. (2020). All settings fine-tune the classifier head, with the following variants: (1) **Head** - classifier head only, (2) **BB-Last** - last backbone layer + head, (3) **BB-Last-3** - last three backbone layers + head, and (4) **BB-1** - first backbone layer + head. Results are presented in Tab. 4. The result indicates that VIRDA consistently outperforms partial backbone finetuning in terms of both classification accuracy by 3-5% and parameter efficiency (only requires 1.5M training parameters versus 8.2M training parameters from BB-1 and BB-Last.

**Effect of Losses.** Tab. 5 illustrates how different combinations of loss functions affect the accuracy of A→D task on the Office-31 dataset. While the normal combination of supervised loss and adversarial loss would be beneficial standard UDA training, for VirDA using this combination decreased the overall accuracy by $-2\%$. We hypothesize that, while $\mathcal{L}_{adv}$ enforces a similar distribution of hidden features, for VirDA this translates to optimizing $f_{mask}$ and the textural features, and also might forces the classifier head to learn more as optimizing the backbone becomes less flexible (since our later Error Analysis discussion indicated that the current $f_{mask}$ might be insufficient in some cases. This can result in overfitting or learning spurious features in the classifier head. By adding a more enforcing uncertainty loss, we force the classifier head also to match the prediction uncertainty, thus minimizing the chance of overfitting. Introducing the inter-domain alignment loss, $\mathcal{L}_{unc}$, significantly improves accuracy to 93.8%, demonstrating its effectiveness in narrowing

the domain gap. Employing a single intra-domain alignment signal, either $\mathcal{L}_{unsup}$ or $\mathcal{L}_{distrib}$, yields marginal improvements of $+3.1\%$ and $+1.9\%$, respectively. However, combining both intra-domain alignment losses enhances performance notably by $+4.1\%$, resulting in the highest accuracy achieved in this task.

Table 5: Incremental accuracy gains for each additional loss term on A→D task on Office-31. The best results are in **bold**.

| Added Loss | Included Losses | | $\Delta$ Acc. (%) | Total Acc. (%) |
|---|---|---|---|---|
| Source-only | – | | – | 90.6 |
| $\mathcal{L}_{sup}$ | $\mathcal{L}_{sup}$ | | +1.6 | 92.2 |
| $\mathcal{L}_{adv}$ | $\mathcal{L}_{sup} + \mathcal{L}_{adv}$ | | −2.0 | 90.2 |
| $\mathcal{L}_{unc}$ | $\mathcal{L}_{sup} + \mathcal{L}_{adv} + \mathcal{L}_{unc}$ | $(\mathcal{L}_{inter})$ | +3.6 | 93.8 |
| $\mathcal{L}_{unsup}$ | $\mathcal{L}_{inter} + \mathcal{L}_{unsup}$ | | +3.1 | 96.9 |
| $\mathcal{L}_{distrib}$ | $\mathcal{L}_{inter} + \mathcal{L}_{distrib}$ | | +1.9 | 95.7 |
| $\mathcal{L}_{intra}$ | $\mathcal{L}_{inter} + \mathcal{L}_{unsup} + \mathcal{L}_{distrib}$ | $(\mathcal{L}_{inter} + \mathcal{L}_{intra})$ | +4.1 | **97.9** |

**Effect of Structural mask-producing layer.** We evaluate the effectiveness of $f_{coord}$ on the same 3 tasks in Office-31 dataset in Tab.6. With both backbones, which are ResNet50 and ViT, adding $f_{coord}$ yields a gain on all three tasks. This highlights that our coordination module scales especially well with transformer backbones, offering clear, consistent gains on the hardest transfers.

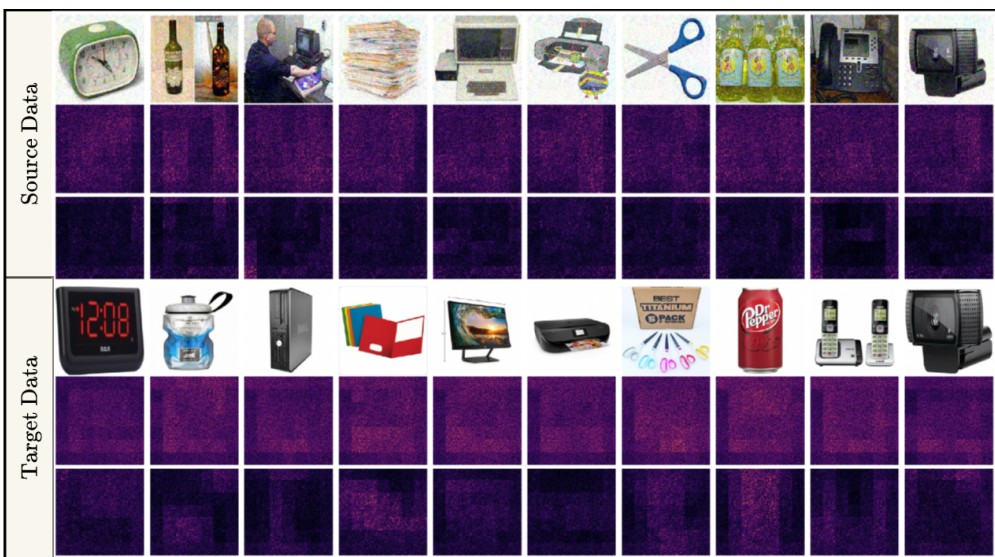

Figure 4: Visualization of the original image, the reprogrammed mask before (upper row) and after (lower row) UDA task on Rw→Pr. The source domain masks focus on encoding the surrounding areas, while the target domain masks highlight the main object.

**Efficiency metrics of VirDA.** We report FLOPs, per-step training time, and peak GPU memory for VIRDA paired with different backbones in Tab. 7. VIRDA cuts training compute by 31–33% GFLOPs and shortens per-epoch time by ≈30% compared to full fine-tuning. Since the VR modules are placed before the frozen backbone, gradients must still traverse the backbone to train these layers. Therefore, the memory usage yields different memory behavior across backbones: for ResNet, where peak usage is dominated by feature maps, freezing changes little; for ViT, eliminating attention-specific weight-gradient buffers and optimizer state reduces memory by up to 25%. In terms of inference computation efficiency, VIRDA introduced a minimal over head of less than 5% increased GFLOPs.

Table 6: Ablation on Office-31 with/without the structural mask-producing layer $f_{coord}$. The best results are in **bold**.

| Backbone | $f_{\text{coord}}$ | $\mathbf{D \to A}$ | $\mathbf{W \to A}$ | $\mathbf{A \to W}$ |
|---|---|---|---|---|
| ResNet50 | $\times$ | 63.2 | 62.3 | 83.1 |
| ResNet50 | $\checkmark$ | 64.2 | 66.7 | 84.9 |
| ViT-B/32 | $\times$ | 74.1 | 81.4 | 90.9 |
| ViT-B/32 | $\checkmark$ | **81.3** | **84.1** | **94.7** |

Table 7: Efficiency metrics of VIRDA: FLOPs, training time, GPU memory usage. Here, **Size** indicate the input size, ↑**GFLOPs** and **GFLOPs (Train)** indicates the added GFLOPs from VIRDA and training GFLOPs accordingly, **Training (s)** indicates average required time per training step in seconds and **Memory (GiB)** indicates the memory usage.

| Method | Size | ↑GFLOPs | GFLOPs (Inference) | GFLOPs (Train) | Training (s) | Memory (GiB) |
|---|---|---|---|---|---|---|
| ResNet50 (Full) | 224 | 0.4 | 8.2 | 24.9 | 0.59 | 12.6 |
| ResNet50 (VirDA) | 224 | | 8.6 | 16.7 | 0.42 | 12.6 |
| ViT-B/32 (Full) | 384 | 1.2 | 25.3 | 77.2 | 0.53 | 24.1 |
| ViT-B/32 (VirDA) | 384 | | 26.5 | 51.8 | 0.39 | 18.2 |
| CLIP (VirDA) | 224 | 0.4 | 1,180 | 1,180 | 0.53 | 12.6 |

**Visualization.** Figure 4 visualizes reprogrammed masks before and after training on source and target samples. Initially (upper rows), the masks exhibit diffuse, unclear patterns. Post-adaptation, source masks emphasize background regions, while target masks focus on modifying primary objects. Masks effectively alter simpler objects like "Soda" or "Telephone", but face challenges with visually different objects like "Computer" or "Printer", and multiple-object scenarios such as "Scissors".

**Error analysis.** Figure 3 visualizes how VIRDA transfers from the Product source to two targets: a mild shift (Realworld) and a hard shift (Clipart). The hardest failures in Clipart (e.g., "Post-it Notes", "Marker", "Eraser") arise when the target removes the photo-like appearance cues, shading, specular highlights, and fine material texture that the visual prompts leverage. The learned reprogram mask then attends to background or non-discriminative regions; this can still reduce the inter-domain discrepancy loss, but does not support correct classification. In the easier Realworld target, where appearance is closer to the source, the prompts highlight object bodies (e.g., marker barrel, eraser edges/print) and accuracy improves substantially. Conversely, classes with distinctive, domain-invariant silhouettes (e.g., "Sneakers", "Flowers", "Flip-flops") transfer well to both targets, even when texture is simplified.

## 5 Conclusion and Future Work

In this paper, we propose a novel method, VIRDA, a parameter-efficient solution for UDA that's capable of reusing a single pretrained backbone for all transfer settings. By introducing lightweight, domain-specific visual reprogramming layers that prepend to the frozen backbone, VIRDA adapts source knowledge to target domains through texture-level transformations rather than full network fine-tuning. We add intra- and inter-domain losses to guide the reprogramming function under the unsupervised constraint. Moreover, we leverage the prediction uncertainty to stabilize the training procedure. Our experiments demonstrate that VIRDA achieves competitive or superior performance compared to prior methods, and approaches the performance of state-of-the-art approaches while using only a fraction of the parameters. In the future, we plan to implement our VIRDA to tackle the challenging downstream tasks, e.g., semantic segmentation and object detection.

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
