# OpenReview forum: "VirDA: Reusing Backbone for Unsupervised Domain Adaptation with Visual Reprogramming"
_TMLR — Accepted by TMLR_

### Review · Reviewer_KoCp · 2025-10-25

**Summary Of Contributions:**

The paper introduces a method for unsupervised domain adaptation (UDA), focusing on textural differences between images from the source domain and images from the target domain. The method learns a lightweight adapter that directly operates on input images. The adapter maps images into the source domain by applying coordinate attention and textural masking. Then, the image is encoded by a frozen backbone (for example, a trained ViT), and class labels are predicted by a domain-specific classifier. The adapter and classifier are optimized through a series of losses: A supervised classification loss on the source domain, a discriminator loss that encourages source and target classifier predictions to be indistinguishable, a loss that aligns uncertainty in classifications on the source and target domains, a divergence loss that aligns predictions on the target domain when applying image transformations, and finally a classification loss on the target domain using pseudo-labels obtained from the source domain classifier.

The proposed method is evaluated against several baselines on three tasks (digits, office-31, and office-home), all standard in UDA. Ablation studies further analyze the effect of different losses and the performance when using different backbones.

**Audience:**

Yes

**Audience Explanation:**

UDA is a well-established field in computer vision, and therefore, many will be interested in developments within this field. The present paper is particularly focused on textural shifts and memory efficiency, both of which are relevant subfields with a significant range of applications, so not achieving state-of-the-art benchmark results is less of an issue.

**Broader Impact Concerns:**

No broader impact concerns. Broader Impact Statement not necessary.

**Claims And Evidence:**

Yes

**Claims Explanation:**

Overall, the paper is technically sound and contains most information to understand the proposed method. In particular
 * The method is evaluated on standard UDA benchmarks
 * It is compared to an extensive set of baselines
 * Results are good, while not state-of-the-art. This is appropriately discussed in the context of contrasting parameter-efficiency and performance. While more expensive methods can achieve better performance, the paper makes a convincing argument regarding the memory efficiency of the proposed method, which could make it an interesting candidate in applications where memory is a concern
 * Ablations are relevant and insightful.
 * The visuals throughout the paper are helpful in understanding the method and results.

Regarding the experimental setup, my main concern is that both digits and office-31 datasets are close to saturation, so it is difficult to assess the significance of any performance differences. This is not a point that needs to be fixed in this paper, but future work may consider switching to more challenging benchmarks.

Finally, there are some issues with clarity which I will discuss below.

**Requested Changes:**

There are main points that require changes:

**(R1)** _Limitations_: While the focus on textural shifts and the memory-efficient scenario is implicit throughout the paper and mentioned occasionally, the paper would benefit from an explicit limitations section where the envisioned range of applications of the proposed method, and its respective advantages and disadvantages, are systematically discussed. This would help readers better contextualize the proposed method in the field of UDA and could inform future developments.

**(R2)** _Clarity_: Because of the many architectural components and losses, the method description is hard to follow. Also, some details are omitted whose inclusion would facilitate understanding. I noticed the following points, which should be addressed in a revision:
  * Motivation of mask in Sec. 3.1. From my understanding, the role of the mask is to select where to apply the domain-specific textural pattern. However, this is not mentioned explicitly. Consider adding are more explicit discussion of the mask's purpose.
  * Sec. 3.1 introduces a channel-last notation ($w \times h \times c$) but the paragraph "Structural mask-producing Layer" assumes $c \times h \times w$, i.e. the reverse. Consolidate these formal disagreements.
  * Also, I think it is helpful in the paragraph "Structural mask-producing Layer" to make the construction of coordinate attention explicit, instead of only referencing Hou et al. (2021). This will make the paper more self-sufficient and make the method easier to understand.
  * Eq. 9 is detached from the corresponding explanation, which is earlier (above Eq. 8). Consider restructuring this paragraph to more tightly couple explanations and formalization.
  * "threshold-based uncertainty filtering" is unclear to me. I understand that the method simply takes the predictions from the source domain classifier (i.e., the most likely label) as pseudo-labels for the target domain classifier. I do not understand the connection to uncertainty and filtering. Please clarify this point.
  * The most important point is to give a high-level overview of the method and losses at the beginning of Sec. 3. This will make the description easier to follow, given the complexity and high number of components. The division into architecture and loss terms is already helpful, and having an overview in the beginning that mentions all components, their roles, and possible relations to each other will increase clarity further.

Finally, there are at least two sentences that require rephrasing:
  * "A mask-producing layer $f_\text{mask}$ is a function that takes as input an image and produces the mask, is a fully convolutional subnetwork of": The double "is a" is confusing and likely not grammatically correct.
  * "and weak augmentations, which are the original image": I understand from the context that, in this case, no augmentations are performed. However, equating this with weak augmentations is confusing.

---

> ### Author Response · Authors · 2025-11-06
> **Paper revision for better clarity**
>
> We thank the reviewer for constructive comments. We have addressed your comments and revised our submission accordingly:
>
> *(R1): Discussion on limitations method's advantage and disadvantages:*
>
> Thank you, we agree that an explicit discussion outlining both the advantages and limitations of our method would help improve the clarity and applicability of VirDA. To address this, we have added a dedicated Discussion section (Section 4.4 Ablations studies / Error analysis) in the revised manuscript. In this section, we systematically analyze the scenarios where VirDA performs well (e.g., $Pr \to Rw$) and underperforms (e.g. $Pr \to Cl$). We also provide qualitative visualizations to illustrate these failure patterns and class-wise analysis.  We conclude that VirDA works best when transferring classes with distinctive, domain-centric textural features, such as shading, specular highlights, and fine material texture that are exploitable by the visual prompts. When these exploitable features are missing, $f_{\text{mask}}$ would instead focus on reprogramming spurious regions, leading to sub-optimal results.
>
> *(R2.1): Making the motivation on the mask of Section 3.1. clearer*
>
> We agree that adding the role of the mask would further improve the clarity of the paper. In Section 3.1, we have added the following sentences to clarify the role of the mask-producer module:
>
> > Intuitively, each visual prompt is a combination of structural masking and textural reprogramming. Hypothetically, the structural mask, i.e., the successive application of $f_{coord}$ and $f_{mask}$, highlights the regions to be re-programmed for each image, while the textural bias should capture the domain-specific style that the backbone has biases towards, following the finding of the existing work (Geirhos et al., 2018).
>
> *(R2.2): Making the construction of coordinate attention explicit in paragraph "Structural mask-producing Layer" to make the paper more self-sufficient*
>
> We agree that adding an explicit formulation of the coordinate attention would make the paper more self-contained. Thus, in Section 3.1. Encoding Domain-specific Textural and Transformational Visual Prompt, we have added (1) the description for Coordinate Attention (CA) and (2) how $f_{coord}$ is constructed from CA.
>
> *(R2.3.) Eq. 9 is detached from the corresponding explanation, which is earlier (above Eq. 8). Consider restructuring this paragraph to more tightly couple explanations and formalization.*
>
> Thank you, we have repositioned EQ.9 to appear before the explanation for clarity
>
> *(R2.4.) "threshold-based uncertainty filtering" is unclear.*
>
> Thank you, we agree that the sentence is over-complicated and adjusted it in our revision:
> > We use the most confident predictions from the weak augmentation branch for each sample, $\hat{y}\_i^{(t)}= \arg \max p^{(t)}\_{i,weak}$, to construct our unsupervised loss:
> $$\mathcal{L}\_{unsup} = \frac{1}{k} \sum\_{i=1}^{k} \mathbf{CE} (\hat{y}\_i^{(t)}, p^{(t)}\_{i,strong})$$
>
> *(R2.5.) Adding a high-level overview of the method and losses at the beginning of Sec. 3*
>
> We agree that adding this loss description will improve the clarity of the paper. Thus, we added a more detailed description of the loss function used in the Overview of Section 3:
> > To optimize this visual reprogramming layer, we follow the formulation of existing works and attempt to align the hidden features produced by applying these layers in Section 3.2. (See Figure 2). These hidden features are aligned intra-domain with domain-specific data augmentations, and inter-domain by aligning hidden features produced with different domain-specific visual reprogramming layers. Specifically, the inter-domain alignment loss is implemented using three sub-loss functions: the supervised source loss $\mathcal{L}\_{sup}$, the adversarial loss that aims to align inter-domain feature alignment $\mathcal{L}\_{adv}$, and $\mathcal{L}\_{unc}$ that aligns between the class-wise prediction uncertainty of source and target domains. The intra-domain loss, meanwhile, is constructed using two sub-loss functions: (1) $\mathcal{L}\_{unsup}$, an unsupervised consistency loss enforcing the same image under two different augmentations yields the same output, and (2) a confidence-distribution matching loss $\mathcal{L}\_{distrib} that encourages the output confidence distributions from differently augmented views of the same image to be similar.
>
> Finally, we have also rephrased the two mentioned sentences:
> > A mask-producing layer $f_{mask}$ is a function that takes as input an image and produces the mask. The function is a fully convolutional subnetwork of $L_{vr}$ layers, where each layer $l \in \{1,..., L_{vr} \}$ performs a $3 \times 3$ convolution with padding of $1$.
>
> > This is done by augmenting the target domain image through both strong augmentations, such as affine transformation or color jitter, and weak augmentations, which are the normalization of original image.

---

### Review · Reviewer_RVu7 · 2025-10-28

**Summary Of Contributions:**

The paper proposes VirDA (Visually reprogrammed Domain Adaptation), a novel and parameter-efficient framework for unsupervised domain adaptation (UDA) in image classification. Instead of fine-tuning the entire backbone for each new domain, VirDA introduces a lightweight, domain-specific visual reprogramming layer that modifies the input images through learnable texture- and structure-aware visual prompts, enabling adaptation to new domains without altering backbone parameters. The method incorporates both inter-domain and intra-domain alignment objectives: the former aligns source and target domain features using supervised, adversarial, and uncertainty-based losses, while the latter refines the target classifier through pseudo-labeling and consistency regularization.

The proposal has been experimentally evalauted on multiple benchmarks (Digits, Office-31, and Office-Home). The strengths of this work lie in its high parameter efficiency, backbone reusability, and empirical results across different datasets and backbones. However, its weaknesses include limited analysis of accuracy that the propose method achieves, and the increased methodological complexity due to multiple loss functions and modules.

**Audience:**

Yes

**Audience Explanation:**

The paper addresses a relevant and timely problem in computer vision—how to perform unsupervised domain adaptation efficiently without retraining large backbones for every new domain. Given the growing interest in parameter-efficient fine-tuning, visual prompting, and resource-efficient transfer learning, many readers of TMLR, especially those working on domain adaptation, transfer learning, and efficient model reuse, would likely find the core idea of visual reprogramming for UDA of interest.

**Broader Impact Concerns:**

While the paper does not raise major ethical concerns, a few broader impact aspects should be discussed explicitly. The proposed method, VirDA, enables efficient reuse of pretrained vision backbones across domains, which could significantly lower the computational and environmental costs of model adaptation. However, by leveraging pretrained models and domain-specific texture reprogramming, the approach may also inherit or amplify existing biases present in the source backbone or datasets used during pretraining. Since the method adapts models without retraining or rebalancing underlying representations, biased visual priors could propagate to new domains, particularly in sensitive applications such as medical imaging or surveillance.

Additionally, the technique’s capacity to repurpose pretrained models across arbitrary visual domains raises potential misuse risks, such as adapting powerful backbones to privacy-invasive or discriminatory tasks without accountability. The paper should therefore include a short Broader Impact section acknowledging (1) the environmental benefit of reducing retraining cost, (2) the risk of bias propagation from reused backbones, and (3) the need for responsible application and dataset auditing when deploying VirDA in real-world settings.

**Claims And Evidence:**

No

**Claims Explanation:**

While the paper presents interesting ideas and reports competitive accuracy, several claims are not fully supported by convincing or clearly justified evidence. First, although the authors emphasize parameter efficiency, Figure 1 suggests that both the forward and backward passes are still computed over the entire network, meaning that the savings in “trainable parameters” may not accurately reflect actual computational or training efficiency. Metrics such as FLOPs, training time, or memory usage would be more appropriate to substantiate these claims. Second, Table 1 shows a notable 10% performance drop on the S→M transfer without any explanation or analysis, raising concerns about the method’s robustness and consistency across domains. Third, the evaluation relies primarily on accuracy and trainable parameter count, but omits more informative measures of efficiency. Finally, certain results appear insufficiently justified—for example, in Table 3, DAPL achieves higher accuracy (70.7% vs. 61.4%) on Pr→Cl with far fewer parameters (0.3 M vs. 1.5 M), which undermines the claimed superiority of VirDA in efficiency–accuracy trade-off. Overall, while the results are promising, the evidence provided does not fully support the strength of the paper’s claims.

**Requested Changes:**

1. The paper’s main motivation is parameter efficiency, yet Figure 1 and the method description imply that the forward and backward passes still traverse the entire backbone. The authors should clarify what computational savings are actually achieved (e.g., fewer trainable parameters vs. fewer FLOPs or reduced training time) and report additional metrics such as FLOPs, wall-clock training time, GPU memory usage, or energy cost. Without this, the claimed efficiency benefit remains unclear.

2. Table 1 shows a ∼10 % drop in accuracy for the S → M transfer task, but the paper provides no explanation or ablation to analyze this behavior. A discussion of potential causes is needed to establish the robustness of the approach.

3. Some results appear inconsistent with the efficiency–accuracy trade-off argument. For example, in Table 3, DAPL achieves higher accuracy on Pr → Cl with only 0.3 M parameters, whereas VirDA uses 1.5 M and performs worse. The paper should analyze why VirDA underperforms or whether differences in backbone settings or optimization explain the discrepancy.

4. The training configuration and implementation details for the visual reprogramming layers (e.g., initialization of texture patterns, mask resolution, hyperparameters for the uncertainty loss) should be fully specified. Code or pseudocode would substantially improve clarity and reproducibility.

---

> ### Author Response · Authors · 2025-11-06
> **Paper revision with additional experiments and discussions on efficiency and limitations.**
>
> Thank you for your detailed comments. We have revised our submission accordingly:
>
> *R1: The authors should clarify what computational savings are actually achieved (e.g., fewer trainable parameters vs. fewer FLOPs or reduced training time) and report additional metrics such as FLOPs, wall-clock training time, GPU memory usage, or energy cost.*
>
> We agree that clarifying the notion of efficiency is important for the positioning of the paper. While the main motivation lies in parameter efficiency (i.e., enabling domain adaptation without retraining or duplicating the backbone for each new source–target pair), we agree that discussing downstream effect on memory, training and inference efficiency would improve the clarity of the paper. Thus, we have added these related efficiency measurements in Section 4.4, and summarized the downstream effects of VirDA below:
> - **Improved Storage efficiency**: In multi-domain adaptation settings, we only need to store one reusable frozen backbone, along with pairs of lightweight domain-specific VR layers instead of full copies of fine-tuned backbones for each domain pair.
> - **Improved Memory efficiency**: As we only need to fine-tune the visual reprogramming layers, we do not need to compute the gradient storage for backbone layers during training. This is our added experiments in Table 7, Section 4.4, VirDA saved up to 25.0% of training memory footprint for the Vision Transformer (ViT) backbone.
> - **Improved training time and computation efficiency**: Correspondingly, we also recorded 30% per-step training time reduction  (0.39 secs vs 0.53 secs) and 30-35% reduction in training GFLOPs  (16.7 to 24.9 for ResNet, and 51.8 to 77.2 for ViT backbone) in comparison with fine-tuning the full backbone.
> - **Minimal overhead in terms of computational efficiency**: In terms of inference computational efficiency, VirDA introduced a minimal overhead (less than 5\% increased GFLOPs) due to added visual reprogramming layers.
>
> *R.2. Discussion on a ∼10 % drop in accuracy for the S → M transfer task in Table 1*
>
> Thank you for pointing this out. In our revision, we have identified that this drop occurred because the original experiments for that transfer did not use the updated backbone structure with the new CoordNet module. After fixing this mistake, we recovered the accuracy by ~7% (to within ~3% of the baseline). We noted this change in Table 1. We also uploaded our trained model and our replication package for reproducibility.
> To explain why this improvement was possible, we defer the discussion to Table 6’s discussion on the Ablation study of the effect of $f_{coord}$. Specifically, in S $\to$ M  task, while it is clear that there are distinct textural features (MNIST is grayscale while SVHN is colored) to be exploited with the visual reprogramming layer, the mask that dictates where to reprogram was insufficient. Using coordinate convolution allows us to have better masks, hence, better results.
>
> *R.3. The paper should analyze why VirDA underperforms or whether differences in backbone settings or optimization explain the discrepancy.*
>
> We agree that discussing the varying performance of VirDA with respect to different backbones and optimization settings would improve the clarity of the paper. While VirDA underperforms in comparison with DAPL on the Pr $\to$ Cl (Product to ClipArt) task in Office-Home (Table 3), it outperforms DAPL on Office-31 across 4/6 tasks and yields a higher overall mean accuracy (Table 2).
> We attribute this difference in performance between different tasks in our general error analysis section in Section 4.4: the ClipArt (Cl) domain from OfficeHome lacks distinctive textural features, requiring models to use more shape-based features (See our added Figure 4.). This is the weak point of VirDA: even though $f_{coord}$ improved the mask generation process, $f_{mask}$ still cannot provide a sufficiently good mask that focuses on shapes in the ClipArt domain. In contrast, for tasks that VirDA outperforms DAPL, e.g., Pr $\to$ Rw (Product to Realworld), or D → W (DSLR camera to webcam), VirDA can readily exploit the textural features for domain adaptation.
>
> *R.4. The training configuration and implementation details for the visual reprogramming layers (e.g., initialization of texture patterns, mask resolution, hyperparameters for the uncertainty loss) should be fully specified.*
>
> Thank you, we fully agree that such details are critical for clarity and reproducibility. To support replication, we have added the supplementary material, including the source code, along with all the corresponding procedures for initializing textural patterns, input normalization, and hyperparameter settings. We have also provided an instruction file (README) that contains step-by-step instructions for reproducing each of our experiments, and have also uploaded our trained model to HuggingFace.

---

### Review · Reviewer_kVZm · 2025-10-29

**Summary Of Contributions:**

- The authors study the usage of visual reprogramming (VR) layers for unsupervised domain adaptation on the task of image classification. Changing the input with the VR layer, the backbone can be kept frozen and due to the authors it is then sufficient to only adapt the classifier (in contrast to adapt the entire backbone). The proposed framework is highlighted as being especially efficient in the number of updated/ learned parameters as the VR layer is relatively small compared to the Backbone.

- 5 losses are assembled to learn the parameters of the VR layer and the classifier layer.


- The authors evaluate their proposed framework on the Digits dataset with a ResNet backbone and on Office-31 and Office-Home dataset with a ViT-B 32 Architecture.

**Audience:**

Yes

**Audience Explanation:**

The bigger the backbones get, probably the more people are interested in having to store only a small part of it to adapt it to different domains. Furthermore, it is computationally cheaper to train which is given the growing size of backbones, also an important aspect.

**Broader Impact Concerns:**

None.

**Claims And Evidence:**

No

**Claims Explanation:**

In my understanding, the main claim is, that the usage of VR layer is especially parameter efficient while retaining a good performance in UDA. However, what is really lacking in my opinion, is an ablation comparing against different baselines using the same combination of the 5 losses:
 - No VR layer, only adapting the classification layer (as a baseline)
 - No VR layer but unfreezing different parts of the backbone (for example a few layers spread  throughout the network, only in the beginning or only in the end of the network).
 - For ViTs the literature for parameter efficient fine-tuning (PEFT) is very large (e.g. LoRA etc.). In my understanding many of these PEFT methods could be also used (learned with the loss), and serve as a baseline to compete with.

For me the paper insists too much that for previous UDA method there is “only” a full-finetuning possible (“the transfer requires fine-tuning both the welltrained classifier along with fine-tuning the backbone”). However, keeping the backbone frozen and adapting only the classifier or adapting intermediate layers (e.g. like in TENT [1]) is also possible, and in my understanding many deep learning methods for UDA are quite unspecific about which weights in a network are updated.
The indeed interesting question if VR layer are better suited than e.g. unfreezing some layers of the backbone, deserves in my opinion a good portion of ablation experiments.

Besides that, are Table 2 and Table 3 very difficult to interpret, mainly because different backbones are used, while the VirDA is only reported for a ViT (which has to be guessed by the parameter size). To keep the generality of the claim it is in my opinion important to verify the hypotheses for different backbones, especially as we see in Table 5, that parts of the framework work especially good with ViTs and maybe a bit less good with the ResNet. Also, comparisons across backbones are in my opinion always quite difficult, as the source-only performance can be quite different.

I am also a bit puzzled about the usage of the 5 losses. Are these losses weighted or scaled ? (if yes, how were these weights/ scale factors set ?). Why is the adversarial loss kept, when we see in the ablation in Table 4, that it has a negative effect ? (does it have a positive effect with the “intra” loss ?) Is the assembling of the 5 different losses only working in the combination with the VR layer+ classifier update, or is this also working when fine-tuning the entire backbone.

While the other concerns are in my opinion very crucial, I see the following one as a bit of a general concern with many UDA frameworks (and probably never perfectly solvable). The presented framework is evaluated on one modality (Images) and a single task (classification). While this is an important pair, the question can always be raised if the presented framework is specific to this or generalizes well to others. In my opinion it would make the paper much stronger, if there are also different tasks or modalities used for evaluation (e.g. semantic segmentation as a task, or point clouds as modality). Otherwise, I would make it at least clear in the claim, that it for now is only covering this single pair (see also my comment in the “minors” of the requested changes sections).

[1]  TENT: FULLY TEST-TIME ADAPTATION BY ENTROPY MINIMIZATION, ICLR 2021.

**Requested Changes:**

**Crucial:**

- Comparing to different baselines where the loss is the same, but different parts of the backbone are unfrozen (early layers, layers spread across the network, later layers, classifier, or other PEFT strategies)

- Clarification about the raised questions of the losses.

- Reporting the performance of VirDA with a ResNet backbone/ CLIP backbone in Table 2 or Table 3.

**Making the paper stronger:**
- Experiment(s) with different modalities or tasks.

**Minors:**
l.1, Page 1:
“Image classification is the foundation of nearly all computer-vision pipelines.”:
While I agree on the importance of image classification, I would suggest to lower the tone a bit, as there are many other tasks: image recognition, semantic segmentation, instance segmentation etc. for also many different modalities e.g. image, video, point clouds. Therefore the “nearly all computer-vision pipelines” claim seems to be at least debatable.

**Typos:**
Page 9 :
In the paragraph “Results on Office-31” there is a typo: “only 1.5 parameters” → “only 1.5 M. parameters”

Sources: Several sources are linked with shortings.
- Bai2024a and Bai2024b are referencing the same source

- Du2024a and Du2024b are referencing the same source

- Ge2022a and Ge2022b are referencing the same source (in Ge2022b is also a typo in the authors first name)

- Yang2023a and Yang23b are referencing the same source

- Zhu2023a and Zhu2023b are referencing the same source

---

> ### Author Response · Authors · 2025-11-06
> **Paper revision on experiments, discussion and artifacts**
>
> Thank you for your comments. We have revised our submission accordingly.
>
> *R1: Comparing to different baselines where the loss is the same, but different parts of the backbone are unfrozen (early layers, layers spread across the network, later layers, classifier, or other PEFT strategies)*
>
> Thank you, we agree adding and discussing these on partial finetuning would improve the positioning of the paper. Thus, we have added the following discussion and Table 4 in the Ablation study (Section 4.4), where we compared VirDA with the following partial finetuning settings:
> > (1) **Head** - fine-tuning classifier head only, (2) **BB-Last** - last backbone layer + head, (3) **BB-Last-3** - last three backbone layers + head, and (4) **BB-1** - first backbone layer + head.
>
> The result indicates that VirDA consistently outperforms partial backbone finetuning in terms of both classification accuracy by 3-5\% and parameter efficiency (only requires 1.5M training parameters versus 8.2M training parameters from BB-1 and BB-Last.
>
> *R2: Table 2 and Table 3 are difficult to interpret, mainly because different backbones are used, while the VirDA is only reported for a ViT (which has to be guessed by the parameter size).*
>
> Thank you, we have restructured Table 2 and Table 3 and grouped the baselines by the used backbones (CNN, ViT, and CLIP). To make the table more comprehensive, we have added the result of combining VirDA with ResNet and CLIP. Finally, we have also updated the two tables’ descriptions and analysis accordingly.
>
> *R3: Verifying hypotheses on why VirDA on ResNet achieved lower accuracy than ViT and discussing backbone compatibility.*
>
> Thank you, we have re-investigated Table 5 (now Table 6), and identified that this performance drop on ResNet occurred because we missed the CoordNet layer in the original ablation experiments. After correcting this mistake, we observed a consistent improvement on both backbones (highlighted in Table 6). To support reproducibility, we have provided the reproduction package with detailed running instructions along with our model checkpoints stored on Huggingface.
>
> *R4. Discussion on (1) weighting/scaling loss application, (2) the effect of the adversarial loss in Table 4. and (3) whether the loss is applicable to other finetuning methods.*
>
> Thank you, for describing how the losses are combined, in Section 3.3 (page 8, equation 13) and Section 4.2 (page 9), we have added details on how these losses are weighted using hyperparameters from grid search. We also made these configurations and the grid search procedure accessible in our replication package.
>
> We discussed the reason behind the negative effect of adversarial loss in Section 4.4:
> > While the normal combination of supervised loss and adversarial loss would be beneficial for standard UDA training, for VirDA, using this combination decreased the overall accuracy by $-$2\%. We hypothesize that, while $\mathcal{L}\_{adv}$ enforces a similar distribution of hidden features, for VirDA this translates to optimizing $f_{mask}$ and the textural features, and also might forces the classifier head to learn more as optimizing the backbone becomes less flexible (since our later Error Analysis discussion indicated that the current $f_{mask}$ might be insufficient in some cases. This can result in overfitting or learning spurious features in the classifier head. By adding a more enforcing uncertainty loss, we force the classifier head also to match the prediction uncertainty, thus minimizing the chance of overfitting.
>
> Finally, while our loss objectives are tailored for VirDA, they are also applicable for other methods (e.g., full backbones or partial backbone finetuning). We have also applied these losses to the results of Table 7.
>
> *R5:  Adding different tasks or modalities, or claiming that the method, for now, is only covering single-pair image classification.*
>
> Thank you. While our settings effectively demonstrate the method’s capability in image classification, we acknowledge that its generalization to other tasks (e.g., semantic segmentation, point clouds, or video) remains to be explored. We have clarified this scope in the revised paper in the future work section.
>
> *R6: Experiment(s) with different modalities or tasks.*
>
> Thank you, we added one variant of VirDA that uses both text and image features with CLIP. In our paper, complementary to VirDA, which tries to align visual representation for domain adaptation, the DAPL baseline instead aligns textual representation. Thus, we created a multi-modal variant of VirDA, namely, VirDA (CLIP), in Tables 2 and 3 by combining both VirDA and DAPL. While this led to improvement in some cases (compared to DAPL), there exist performance drops in others. We believe that coming up with better training strategies to better incorporate both text and image would be an interesting direction to explore.
>
> Finally, we have also fixed all the mentioned typos and removed the duplicate references.

---

### Author Response · Authors · 2025-12-05
**Acknowledgement of Acceptance and Camera-Ready Instructions**

Dear Action Editor and Reviewers,

Thank you for your constructive feedback during the review stages. We are happy to proceed with the publication of our work.

We confirm that we will update the tables in the final camera-ready version to remove the bold formatting from our method and apply it only to the best-performing results, as requested in the decision statement.

Best regards,
The Authors

---

### Decision · Action_Editor_Mo5n · 2025-12-05

**Recommendation:** Accept as is

**Additional Comments:**

Please take into account the comment about bold usage in table for camera ready.

**Audience:**

Yes

**Audience Explanation:**

UDA is a fundamental challenge in machine learning and computer vision and it is definitely of interest to the community.

**Claims And Evidence:**

Yes

**Claims Explanation:**

The paper introduce a lightweight Unsupervised Domain Adaptation method using an adapter on input images. All reviewers found the method interesting but had some required changes including clarifications, computational cost and ablations. The authors did a revision of the paper that addressed most of the reviewers concerns. They all agree that the paper if OK for acceptance at TMLR and I concur.

I recommend to accept the paper "as is" to avoid back and forth but ask that the authors update their tables in the camera ready version by removing bold for their method and use the bold for the best performing (in accuracy or parameter size) and possibly underline for second best as is standard in the community. Being always best is not important but respecting commonly accepted usage of bold in table is.